# Health system barriers to hypertension care in Peru: Rapid assessment to inform organizational-level change

Kendra N. Williams[1,2]*, Janeth Tenorio-Mucha[3], Karina Campos-Blanco[3,4], Lindsay J. Underhill[5], Armando Valdés-Velásquez[4,6], Antonia Fuentes Herbozo[3], Laura K. Beres[1], Lisa de las Fuentes[5], Lucy Cordova-Ascona[3,4], Zoila Vela-Clavo[3,4], Gonzalo Mariano Cuentas-Canal[7], Juan Carlos Mendoza-Velasquez[8], Sonia Mercedes Paredes-Barriga[8], Raquel Hurtado La Rosa[9], Makeda Williams[10], Elvin H. Geng[11], William Checkley[2,12], Joel Gittelsohn[1], Victor G. Davila-Roman[5], Stella M. Hartinger-Peña[3,4]

1 Department of International Health, Social and Behavioral Interventions Program, Bloomberg School of Public Health, Johns Hopkins University, Baltimore, Maryland, United States of America, 2 Center for Global Non-Communicable Disease Research and Training, School of Medicine, Johns Hopkins University, Baltimore, Maryland, United States of America, 3 Facultad de Salud Publica y Administración, Universidad Peruana Cayetano Heredia, Lima, Peru, 4 Centro Latinoamericano de Excelencia en Cambio Climático y Salud, Universidad Peruana Cayetano Heredia, Lima, Peru, 5 Global Health Center, Institute for Public Health and Cardiovascular Division, Department of Medicine, Washington University School of Medicine, St. Louis, Missouri, United States of America, 6 Laboratorio de Estudios en Sistemas Socio-Ecológicos, Universidad Peruana Cayetano Heredia, Lima, Peru, 7 Departamento de Medicina, Servicio de Medicina, Hospital Base III EsSalud, Puno, Peru, 8 Direccion Regional de Salud (DIRESA; Regional Health Ministry), Puno, Peru, 9 Dirección de Prevención y Control de Enfermedades No Transmisibles (Directorate of Prevention and Control of Non-Communicable Diseases), Ministerio de Salud (MINSA; Ministry of Health), Lima, Peru, 10 Center for Translation Research and Implementation Science, National Heart, Lung, and Blood Institute, National Institutes of Health, Bethesda, Maryland, United States of America, 11 Center for Dissemination and Implementation, Institute for Public Health, and Infectious Diseases Division, Department of Medicine, Washington University School of Medicine, St. Louis, Missouri, United States of America, 12 Division of Pulmonary and Critical Care Medicine, School of Medicine, Johns Hopkins University, Baltimore, Maryland, United States of America

* kendra.williams@jhu.edu

**Data Availability Statement:** The data underlying this study cannot be made publicly available due to

## Abstract

Traditional patient- and provider-level hypertension interventions have proven insufficient to halt hypertension as the leading cause of morbidity and mortality globally. Systems-level interventions are required to address factors challenging hypertension control across a social ecological framework, an under-studied topic particularly salient in low- and middle-income countries (LMICs) such as Peru. To inform such interventions, we sought to identify key health systems barriers to hypertension care in Puno, Peru. A participatory stakeholder workshop (October 2021) and 21 in-depth interviews (October 2021—March 2022) were conducted with 55 healthcare professionals (i.e., doctors, nurses, midwives, dentists, nutritionists), followed by a deductive qualitative analysis of transcripts and notes. Participating healthcare providers indicated that low prioritization and lack of national policies for hypertension care have resulted in limited funding and lack of societal-level prevention efforts. Additionally, limited cultural consideration, both in national guidelines as well as by some providers in Puno, results in inadequate care that may not align with local traditions.

compliance with our Institutional Review Board (IRB)-approved protocol and consent form, which ensured participants' anonymity and confidentiality. Given the qualitative nature of the data, participants are highly identifiable, even with anonymization efforts. Public availability would compromise participant privacy and potentially endanger their employment or livelihood, as some participants voiced concerns about the healthcare system. While we are open to sharing our data upon request, access will be granted only under a signed agreement prohibiting further dissemination or sharing of the transcripts. Please contact Peter Dore, Database Administrator, at pmdore@wustl.edu for data access requests.

**Funding:** The study was supported by the National Heart, Lung, and Blood Institute (UG3 HL152371) NIH, Bethesda, MD, USA (Multiple Principal Investigators: Hartinger-Peña, Davila-Roman, Checkley, Geng). We also acknowledge the Global Alliance for Communicable Diseases (GACD), London, UK, the Global Health Center, Institute for Public Health and Department of Medicine, Washington University in St. Louis, and the Research training in chronic, non-communicable respiratory diseases in Peru training grant (D43TW011502; MPIs: Checkley, Hartinger-Peña). EG was also supported by a K24 (AI134413) and an educational grant from Viiv Healthcare. The funders were not involved in the development of the study design or collection, management, analysis or interpretation of data. The program officer representing the National Heart, Lung, and Blood Institute participated in Data and Safety Monitoring Board activities.

**Competing interests:** The authors have declared that no competing interests exist.

Providers highlighted that patient care is also hampered by inadequate distribution and occasional shortages of medications and equipment, as well as a lack of personnel and limited opportunities for training in hypertension. Multiple incompatible health information systems, complicated referral systems, and geographic barriers additionally hinder continuity of care and care seeking. Insights gained from health providers on the healthcare system in Puno provide essential contextual information to inform development of organizational-level strategies necessary to improve provider and patient behaviors to achieve better hypertension care outcomes.

## Introduction

Hypertension is the leading cause of morbidity and mortality worldwide, particularly in low- and middle-income countries (LMICs) [1]. Nearly 1.4 billion people globally have hypertension [2]. If detected early, hypertension can be controlled through lifestyle modifications and adherence to medication regimens before target organ damage, such as heart failure, stroke, and renal disease, manifest. However, rates of hypertension diagnosis, treatment and control are low [3]. In Latin America and the Caribbean, only an estimated 35% of women and 23% of men with hypertension have their blood pressure under control [4]. In Peru, nearly 20,000 premature deaths and 370,000 disability adjusted life years (DALYs) annually are attributed to hypertension [1].

Understanding healthcare system barriers to hypertension care, such as those related to governance and leadership, financing, access to medicines and essential technologies, health personnel, health information systems, and service provision, is necessary to identify strategies that can improve the quality of care and significantly reduce adverse hypertension outcomes [5]. Individual-level interventions targeting medication non-adherence are common, but organizational interventions at the structural, facility and community levels are critical for change and less well understood [6].

Although previous research has investigated barriers to hypertension care and treatment adherence [6, 7], there is little research on health systems barriers faced in indigenous, remote LMIC populations. The Addressing Hypertension and Diabetes through Community-Engaged Systems (ANDES) trial aims to improve hypertension diagnosis, treatment, and control in an indigenous population living at high-altitude in Puno, Peru through a multi-component intervention leveraging community health worker (CHW)-led implementation strategies. The ANDES trial investigators sought to obtain an in-depth understanding of existing health system barriers to hypertension care as one component of several formative research activities, which together informed the design of an intervention to be tested in a randomized controlled trial.

## Methods

### Study setting

The formative research was carried out in Puno, Peru. Puno is the capital of the Puno Region and the Puno Province (2017 census population: 1,172,697) located in southeastern Peru, at 3,825 meters above sea level, and primarily consisting of indigenous Andean people of Quechua and Aymara origin (population: 537,972 and 318,363, respectively), which together comprise >70% of the population [8]. According to the World Bank, Puno had a poverty rate of

8.3% in 2023, ranking highest out of all regions of Peru [9]. Puno city, the urban provincial capital (138,912 residents) borders Lake Titicaca and northern Bolivia [8]. Rural Puno is comprised of 14 large districts with each district containing up to 20 communities [8]. A previous cross-sectional, population-based study conducted in Puno, Peru (CRONICAS) found that among 1,155 patients aged ≥35 years (n = 574 urban and n = 581 rural; 60% overweight or obese and 53% female), 13.1% had hypertension [10].

## Peruvian health care system

The public health system in Puno is managed by the Regional Health Administration (Dirección Regional de Salud, or DIRESA for its acronym in Spanish), under the Peruvian Ministry of Health (Ministerio de Salud, or MINSA), the governing body that leads, regulates, and promotes national health policy. EsSalud is the government healthcare system that provides health insurance and healthcare to those who are actively employed outside of the home, and their families. As of 2022, most people in Puno (66.6%) receive health insurance through the national governmental insurance (Seguro Integral de Salud, or SIS); 11.2% are covered by EsSalud, 0.6% through other insurance (i.e., armed forces, police, or private insurance), and 21.6% are uninsured [11]. Insurance covers all medical care and medication costs, but in the event of stock-outs, patients must pay out-of-pocket for medications at private pharmacies.

DIRESA Puno is comprised of 11 networks with 454 primary healthcare facilities and 11 general hospitals. EsSalud separately manages nine facilities in Puno. Higher-level health facilities are typically located near city centers, while rural areas have fewer, more dispersed, and lower-level facilities available. Although the Peruvian government has a national program in which medical graduates and healthcare professionals must serve at least one year as a primary care physician in a poor remote area (Servicio Rural y Urbano Marginal de Salud or SERUMS), rural areas of Puno still face severe human resource shortages (1 physician per 689 people in 2020) and challenges with high staff turnover [12]. Traditional healers using medicinal plants and spiritual treatments for various conditions, including heart problems and circulatory system disorders, are also common in the Andean region of Peru [13]. However, recent research suggests high acceptance of modern medicine [14].

Peruvian clinical guidelines specify that the diagnosis and treatment of hypertension take place mainly in primary healthcare facilities by physicians, with referral to higher-level or specialty care for patients with complications [15, 16]. Blood pressure should be measured routinely in all adults over 18 years of age who seek care at a health facility. In health facilities with a physician, people with high blood pressure may be diagnosed and given counseling and medication prescriptions. Facilities without a physician can only provide education on preventive activities and refer the patient to a higher-level facility for diagnosis and treatment. After diagnosis, routine hypertension follow-up care can be provided at lower-level facilities.

Community health workers (CHWs) complement the work of health care professionals by providing health promotion and prevention activities in coordination with healthcare personnel outside of the health facilities; however, their training and activities are usually focused on specific health topics, and they are not currently trained to address chronic disease. According to the MINSA registry, there are around 35,000 CHWs nationwide across Peru, with 900 in the Puno region [12]. CHWs are typically unpaid volunteers elected and recognized by their community to perform this role.

## Study design

We sought to understand health system-level barriers to hypertension care in Puno, Peru. Data collection was done through a rapid assessment, defined as an in-depth, collaborative

qualitative investigation using triangulation to understand a specific situation from the local perspective in a short amount of time [17]. Our rapid approach, conducted during the formative research phase of the ANDES trial, was necessary to ensure findings would inform tailoring of the intervention design. Research data were collected through a participatory stakeholder workshop and in-depth qualitative interviews, as described below.

## Stakeholder workshop

A two-day (October 14–15, 2021) participatory stakeholder workshop was conducted in Puno, Peru with participation from 34 healthcare professionals (i.e., doctors, nurses, midwives, dentists, nutritionists, and other healthcare professionals), representing healthcare facilities and hospitals from 11 provinces of the Puno region (see S1 Table for details). Participants were recruited by a contact in DIRESA who purposefully selected professionals who had leadership responsibilities related to hypertension care, represented various provider types, and came from different regions of Puno. All participants were over the age of 18.

The workshop consisted of two parts. First, participants self-selected to participate in one of six participatory discussion groups (n = 5–6 per group), each addressing barriers within a specific element of the health system including: 1) Governance and leadership, 2) Financing, 3) Access to medicines and essential technologies, 4) Health personnel, 5) Health information systems, and 6) Organization of service provision. Each participatory discussion group was led by a facilitator trained in qualitative research and lasted approximately 60 minutes. Facilitators followed the Spanish version of a rapid assessment guide developed by the Pan American Health Organization (PAHO) as part of the World Health Organization's HEARTS Initiative [18, 19], which aims to improve hypertension disease management in primary care settings [19]. A dedicated note-taker captured key points from each group, which were reviewed and agreed upon by all group members at the conclusion of each discussion.

Second, participants were randomly assigned to groups of five to six members to discuss an example hypertension control intervention theory of change (see S1 Fig). A theory of change depicts how intervention activities are expected to lead to a desired impact based on a logical pathway through intermediate outcomes [20, 21]. The research team presented a draft theory of change featuring three potential intervention components, including: 1) health fairs to bring in community members for blood pressure measurement and screening, 2) capacity building and training activities for healthcare personnel, and 3) a home-based hypertension education and counseling intervention provided by CHWs to improve hypertension control. The theorized proximal outcomes of the intervention were: 1) improved hypertension diagnosis, 2) improved management of hypertension patients by healthcare personnel, and 3) enhanced adherence by patients to healthy lifestyles and medication regimens, with the distal outcome of improved hypertension control. A facilitator presented the theory of change, visualized as a flow diagram hung on a large poster. Participants then discussed it and used post-it notes to record their comments on potential barriers within the current healthcare system that could impede such an intervention (see Fig 1). Each group discussion was facilitated by a trained moderator and lasted approximately 90 minutes.

## In-depth interviews

Four local fieldworkers who were trained in qualitative data collection carried out interviews with 21 healthcare professionals, including 7 nurses (1 in a leadership position), 12 doctors (3 in leadership positions), one midwife (in a leadership position), and one dentist (in a leadership position) (see S2 Table for details). Participants were purposefully recruited from healthcare facilities in rural and urban areas of Puno that ranked highest in number of hypertensive

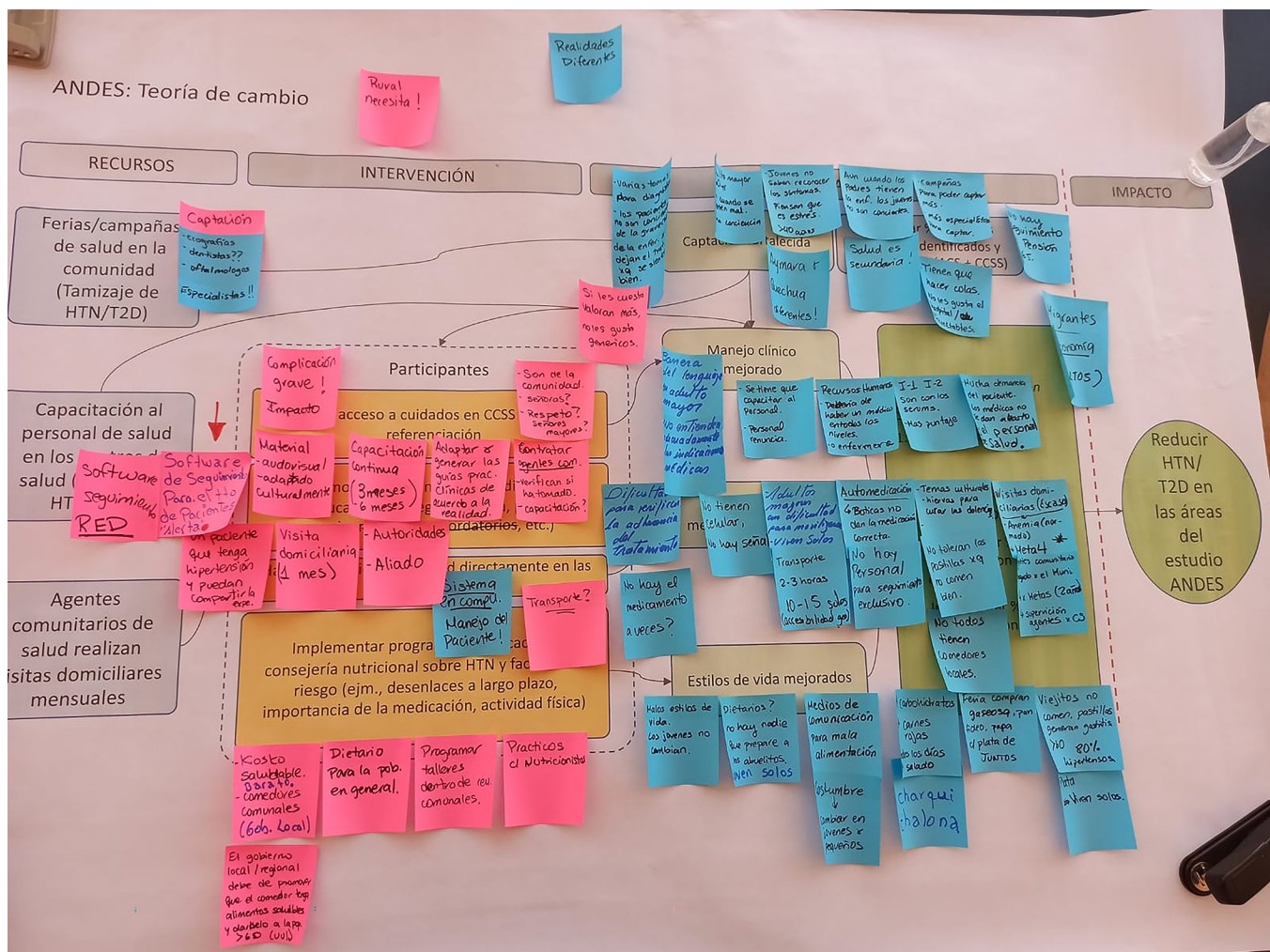

**Fig 1. Example theory of change with group comments.** An English version is available in S2 Fig.

cases reported according to DIRESA data, including both hospitals and lower-level healthcare establishments. We recruited healthcare professionals in those facilities who provided care for hypertensive patients, seeking variation in the characteristics theorized to be associated with outcomes of interest [22], including range of experience (number of years working at the facility), positions of leadership, and type of provider.

Interviews were conducted between November 2021 and March 2022 by trained qualitative researchers. Interviewers used a semi-structured interview guide to discuss topics including the participants' typical employment duties and activities, barriers to providing hypertension care to patients, experience with treating hypertension, and general experience and perceptions of the healthcare system. All interviews were audio recorded and transcribed verbatim in Spanish. An observer also took extensive notes during the interviews, which were expanded to include additional details about topics discussed or observations about the interview within 24 hours.

## Data analysis

We conducted deductive analysis on the data using the six categories from the PAHO rapid assessment guide (i.e., governance and leadership, financing, access to medicines and essential

technologies, health personnel, health information systems, and organization of service provision). Workshop notes, comments on the visualized posters, interview transcripts, and interview notes were coded in Atlas.ti [23] according to the six themes. One coder analyzed the interview materials (AFH), and a separate coder analyzed the workshop materials (KNW); all coded text was reviewed by a small group of authors (KNW, JTM, KCB, AFH, JG) and any disagreements were resolved through dialogue. There were no instances in which a consensus could not be reached. Memos were developed to capture key information related to each theme. All thematically relevant data from any source were combined and analyzed together for each theme. The small author group identified emerging sub-themes by grouping similar quotes together within each theme. Finally, the information was synthesized to develop a summary of key findings within each theme and sub-theme. We sought to establish credibility and confirmability through consistency in data collection (following a clear protocol for research procedures), triangulation (using multiple methods to cross-verify findings and engaging participants with varying roles and locations), and member checking (through which key workshop and interview participants provided feedback and confirmed the findings) [24].

## Ethics

Ethical oversight was provided by the Universidad Peruana Cayetano Heredia in Lima, Peru Research Ethics Committee (277-27-21). The study was also reviewed and approved by the Johns Hopkins School of Public Health Institutional Review Board (IRB00296694) and the Washington University in St. Louis Human Research Protections Office (202108158). Written informed consent was obtained from all participants.

## Results

The rapid health system assessment revealed important barriers to hypertension care within the existing healthcare system in Puno, Peru related to each of the six deductive categories (see summary in Table 1). Findings were consistent across data sources (i.e., both workshop sessions and the in-depth interviews).

### 1. Governance and leadership

**1a. National guidelines are incompatible with the local context.**  Participants noted that national clinical guidelines for hypertension management developed centrally in Lima are often not appropriate for the local context in Puno as they do not consider the cultural background, resource availability, or language variability of the local population (see examples in Fig 1 and S2 Fig–English version).

*"When the nutritionists consult with patients, they give them a diet that they should follow more or less. But this is economically challenging for the patients to do. . . In our area, people eat mostly carbohydrates, potatoes and 'chuño' (dried potatoes). The recommendation to eat more proteins and vegetables is not helpful because patients do not have money to buy them."–Doctor, urban healthcare facility*

**1b. Lack of national-level integration of hypertension prevention into policies.**  Participants stated that there is little value placed on the promotion of healthy behaviors at a societal level and there has been little effort to make rural environments more conducive to healthy lifestyles (including healthy diets and exercise). Some communities have established communal kitchens where community members cook healthy food and eat together, but these were

**Table 1. Key barriers to hypertension care in Puno, Peru according to the six categories of the PAHO rapid health system assessment guide [19].**

| PAHO Themes | Sub-themes identified in Puno | Key barriers identified in Puno |
|---|---|---|
| 1. Governance and leadership | 1a. National guidelines are incompatible with the local context | National guidelines lack consideration of local culture, beliefs, and languages (Quechua and Aymara) |
| | | Guidelines do not account for limited financial and other resources available in Puno |
| | 1b. Hypertension prevention under-valued by leadership and not prioritized in national policies | Lack of societal level health promotion |
| | | Financial assistance through social programs sometimes spent on unhealthy food |
| 2. Financing | 2a. Limited funding for hypertension prevention, diagnosis, and control | Prioritization of maternal-fetal health, childhood anemia, and COVID-19 |
| | | Lack of understanding among policymakers around costs and benefits of hypertension care limits investment |
| 3. Access to medicines and essential technologies | 3a. Challenges with supply and distribution of hypertension medications | Shortages of antihypertensive medications |
| | | Poor demand tracking resulting in medication oversupply at hospitals and undersupply at primary healthcare facilities |
| | | Disorganized distribution system results in delays and medication expiration |
| | | COVID-19 pandemic resulted in less antihypertensive medication usage than normal, presenting challenges for estimating future supply needs |
| | 3b. Lack of equipment and inadequate supplies impede patient care | Limited availability of blood pressure measurement equipment and no maintenance support |
| | | Lack of laboratory facilities and supplies for analyzing bloodwork |
| 4. Health personnel | 4a. Healthcare facilities are understaffed and lack training opportunities | Lack of personnel, especially in primary healthcare facilities in rural areas |
| | | Overworked personnel must focus on emergency medical conditions |
| | | High staff turnover given that personnel are overburdened, underpaid, and lack job security |
| | | Limited training in hypertension and lack of capacity building opportunities |
| | 4b. Challenges resulting from cultural differences between healthcare providers and indigenous populations | Communication barriers given providers typically do not speak indigenous languages |
| | | Lack of knowledge and appreciation of local cultures and traditional medicines |
| | 4c. Community health workers as a potential solution for overburdened healthcare providers | Support from CHWs is currently limited because they are not paid and have little to no training in hypertension |
| 5. Health information systems | 5a. Multiple health information systems are incompatible with each other and poorly maintained | Data from different health information systems are not standardized or integrated |
| | | Some healthcare facilities lack electronic medical records and/or internet access, especially in more remote areas |
| 6. Organization of service provision | 6a. Poorly organized referral systems | Inadequacy of primary healthcare facilities drives use of hospitals for primary care and results in overburdening at hospitals |
| | | Referral guides not updated consistently |
| | 6b. Service organization often not conducive to patient needs | Geographic distance and limited transportation impede care seeking |
| | | Restricted facility service hours sometimes do not align with community needs |
| | | Lack of cell phones and cell service inhibit patient follow-up |

not widespread. Multiple providers stated that raising awareness of healthy eating, healthy lifestyles (i.e., exercise), and awareness about hypertension is needed, for example:

> "*It would be good if there were more diffusion [of health information]. The local and regional health agencies could advertise on the most popular radio stations or programs. . . They could talk about the importance of detecting [hypertension] and then people would be more aware. People would learn that [hypertension] can be treated and then they would come to the health establishments for an exam.*"–Nurse, rural healthcare facility

While financial assistance is given through government social programs such as *Pension 65* (for those 65 and older) and *Juntos* (for poor families with young children), some participants believed that the money is sometimes used to purchase unhealthy, high caloric content foods.

## 2. Financing

**2a. Limited funding for hypertension care.** Many participants stated that maternal-fetal health and childhood anemia are the major healthcare priorities in Puno, receiving the largest portion of the limited government healthcare budget. This leaves little funding for diagnosis and management of hypertension, as well as for hypertension awareness and prevention. Furthermore, due to COVID-19 pandemic-related increases in healthcare expenditures since 2020, hypertension funding was reduced.

*"Currently, vaccinations are the priority. Caring for children with anemia comes second... Since some vaccines will expire soon, [the Ministry of Health] is pressuring us to visit households [to give out vaccinations]. Some health providers go, and as a result, there are fewer personnel at the health facility. Since the Ministry is prioritizing vaccines, they are not asking us about other [health problems], which limits us from following up [on hypertension care]."– Nurse, rural healthcare facility*

Participants mentioned a need for cost-benefit analyses to better quantify the potential benefits from investments in hypertension care. Participants also suggested that having better systems and databases for tracking healthcare spending would facilitate better budgeting and planning and could provide rationale for greater investment in hypertension prevention and treatment.

## 3. Access to medicines and essential technologies

**3a. Challenges with supply and distribution of hypertension medications.** Participants reported that medication shortages in general, particularly those for hypertension, are common in the Puno region, and can lead to medication non-adherence.

*"We schedule [hypertensive] patients to come monthly even though the guidelines state that they should come every three months once their blood pressure is controlled. But we can't give them a three-month supply of [antihypertensive] medication because we don't have enough. Last month, for example, I didn't have enough [medicine] for my patients. I had to submit a request to the health network and they loaned me some from other facilities... Sometimes we don't have [one kind of hypertensive medicine], but we do have [another kind]. We have to give them what we have so at least they can continue with treatment, but we have to change their medical regime... We're in a bad situation with medicines."–Doctor, rural healthcare facility*

Participants also reported frequent oversupply or undersupply of medications, which they attributed to organizational barriers within the medication distribution system. They said hospitals often have an oversupply of medicines (given that hospitals are farther away and harder to access for many patients), while medicines at primary healthcare facilities are lacking. Several participants described that medicines at the hospitals sometimes expire before they are used, suggesting the need for a better system of regularly distributing medicines to primary care facilities in quantities that can be accommodated given limited medication storage capacities. Additionally, participants reported long delays between medication requests and receipt

at healthcare facilities, and that medicines are frequently delivered to healthcare facilities shortly before their expiration dates, exacerbating challenges with the medication supply chain.

Additionally, participants described that healthcare facilities often have trouble estimating how much medicine they will need, as some patients require larger doses than others, some fail to return for follow-up or are not tracked well, and the number of patients can vary from year to year. Some participants stated that during the COVID-19 pandemic, many patients avoided health facilities, which resulted in an oversupply of antihypertensive medications. This situation could lead to medication undersupply in future years as patients return to the health-care system but medication stocks remain at the COVID-19 level usage.

**3b. Lack of equipment impedes patient care.**   According to participants, equipment for blood pressure measurement is often lacking and/or malfunctioning, precluding accurate patient evaluation and/or medication adjustments. Suppliers provide maintenance for equipment during the warranty period, but long-term maintenance is essentially non-existent.

> *"We have an electrocardiogram [at our health facility], but most health establishments in the region don't have one. . . Four healthcare facilities in our network are equipped with electro-cardiogram machines. But if you go to the other facilities right now, only ours works. . . The others are covered in dust. There are many challenges."–Doctor, urban healthcare facility*

Participants also described that many health facilities do not have laboratories for analyzing bloodwork, or they lack supplies for analysis, resulting in incomplete assessments for patients. This causes delays or inability to obtain diagnoses and hinders provision of adequate patient care.

## 4. Health personnel

**4a. Healthcare facilities are understaffed and lack training opportunities.**   Most participants stated that the healthcare system in Puno is severely understaffed, especially in rural areas where health providers are often underpaid and lack job security. As a result, personnel are overworked and unable to meet the healthcare demands of the population. Participants noted that most healthcare professionals must focus on caring for patients with emergency medical conditions, limiting their ability to provide health promotion and/or disease prevention. The limited number of providers also results in long wait times for patients, which discourages patients from seeking care for chronic conditions such as hypertension, and results in limited follow-up of existing patients. Participants also described high staff turnover and a lack of support for managerial and administrative work, which causes challenges with care coordination and follow-up.

> *"[Recent medical school graduates performing mandatory rural healthcare service] only come for one year. Our contracts are annual, it's not like we stay 6 or 7 years. When we arrive, we don't know the population, we don't know how things are here. . ., we don't know who is hypertensive or diabetic because [our health facility] does not have a registry . . . I think this happens in all health facilities where there are recent medical graduates. . . We don't figure out who should be taking medication until our sixth or seventh month, after they should have already been on treatment, and [the patients] lose continuity."–Doctor, rural healthcare facility*

Participants stated that many healthcare professionals have limited training in hypertension. These gaps in knowledge stem from infrequent and/or discontinuous training, limited

access to clinical guidelines and educational materials, and little feedback from superiors in the healthcare network regarding optimal patient care. Capacity building is limited and/or non-existent, mostly because funds are limited and/or trained personnel are not available or not compensated to offer training.

*"There has not been a specific course on hypertension or diabetes. Usually, we get information on hypertension from the [national] guidelines . . . and from the American Heart Association. They update it every two or three years and then we review it. . . There are no courses on [hypertension]. We have to develop this knowledge on our own time."–Doctor, urban health-care facility*

**4b. Cultural differences between health providers and indigenous populations cause challenges.** All participants described communication challenges faced by healthcare providers in treating the predominantly indigenous population in Puno. Healthcare providers speak Spanish but typically do not speak the indigenous languages (Quechua and Aymara), resulting in difficulties communicating with patients, especially elderly patients who do not speak Spanish.

*"There are some patients we can't understand because they only speak Aymara. We have a CHW who acts as an interpreter on Thursdays. Other days, we ask patients to help [trans-late]."–Doctor, rural healthcare facility*

Furthermore, participants described how most healthcare providers lack knowledge of local social and/or traditional culture, including the use of traditional or herbal medicines, leading to healthcare that may be incompatible with traditional beliefs or lifestyles. Participants suggested that this cultural insensitivity could be due to inadequate training and/or limited integration of health providers into the community.

*"People here believe more in traditional medicine. They say, '[The healer] has cracked my fin-gers and back to drive away [the illness]'. . . They come to us with that but we have to explain and orient them [to the medical explanation]."–Nurse, rural healthcare facility*

**4c. Community health workers as a potential solution for overburdened healthcare pro-viders.** Many of the participating providers agreed that use of community health workers to assist with hypertension care, also known as task-shifting responsibilities, could help reduce the burden on providers and improve cultural sensitivity of care. Currently, participants described that support from CHWs is limited because they are not officially employed by the healthcare system and have little or no training in hypertension. Participants stated that healthcare facilities were not able to hire and pay CHWs and, as such, very few are willing and able to volunteer.

## 5. Health information systems

**5a. Health information systems are incompatible and poorly maintained.** According to participants, there are three different and incompatible health care data management systems used in health facilities (the HIS system managed by the national Ministry of Health, the SIS system managed by the national health insurance program, and a registry system managed by the national vital records body). Barriers noted from the use of these different systems within the same health facility included inconsistent data collection and challenges with standardiza-tion and integration. Furthermore, many healthcare facilities have no electronic health records

or lack internet access, requiring the use of paper records that are poorly organized and often missing during follow-up visits. The high staff turnover described above, as well as limited training in record keeping, also impede maintenance of adequate records and can result in under-reporting of hypertension cases.

> "*I think the biggest problem we have in our sector is data. We do not have a system through which we can filter patients and follow-up... Just this morning, a 70-year-old man came in who we had diagnosed in 2019... We lost him [to follow-up]... When he came back, he was not on treatment. He had been buying medicine [over-the-counter], but he hadn't taken anything for the last three months.*"–*Doctor, urban healthcare facility*

### 6. Organization of service provision

**6a. Poorly organized referral systems.** Participants explained how patients' first point of contact with the healthcare system should be at primary healthcare facilities, with referrals to hospitals for more serious problems. However, workshop participants noted that patients often seek initial care from hospitals, or they are referred from primary healthcare facilities to hospitals for the management of uncomplicated hypertension for reasons mentioned above (i.e. inadequate staffing, lack of training, or lack of medication/equipment at primary healthcare facilities). As a result, hospitals can become overloaded, thus limiting their ability to provide adequate attention.

> "*What we do here is if there is a new patient or we detect that they have high blood pressure, we refer them to Puno or Acora [nearby cities with larger health establishments]. They do the detection there, and if they have diabetes or something, the doctor there prescribes them a treatment and we continue it here.*"–*Nurse, rural healthcare facility*

Timely and efficient referrals are often further complicated by the fact that referral guides are not updated regularly and patients often do not complete referrals.

**6b. Service organization often not conducive to patient needs.** Geographic barriers, limited transportation, and restricted facility service hours that do not align with community needs also negatively impact timely access to and compliance with follow-up visits at healthcare facilities, participants stated. Many areas lack cellphone service, and some people lack cellphones, making it challenging for patients to schedule care and for health facilities to contact patients about follow-ups and medication.

> "*There is no cell signal here, even within the health facility. Most patients are older adults who do not have cell phones, or if they have cell phones, they can't receive WhatsApp messages or that kind of thing... People often have to leave their house and go to a specific place where they can receive messages or calls.*"–*Doctor, rural healthcare facility*

### Discussion

We conducted a participatory stakeholder workshop and in-depth interviews with healthcare personnel in Puno, Peru related to hypertension as part of the ANDES trial formative work. We found that low prioritization and lack of national policies for hypertension care has resulted in limited funding and lack of societal-level prevention efforts. Additionally, limited cultural consideration, both in national guidelines and by some providers in Puno, results in

inadequate care that may not align with local traditions. Providers highlighted that patient care is also hampered by inadequate distribution and occasional shortages of medications and equipment, as well as a lack of personnel and limited opportunities for training in hypertension. Multiple incompatible health information systems, complicated referral systems, and geographic barriers additionally hinder continuity of care and care seeking.

The global agenda for health systems strengthening has historically focused mainly on specific infectious diseases such as HIV, tuberculosis, malaria, and vaccine-preventable diseases [25]. In line with this, most existing health systems assessments conducted in LMICs have focused on the ability of primary healthcare settings to address and integrate care for communicable diseases [26]. Also, research on responses to chronic disease often focus on tertiary, curative, and specialist care [25]. Given that noncommunicable diseases, including hypertension, are responsible for 41 million deaths globally each year, 77% of which are in LMICs, there is an urgent need to identify strategies for improving the healthcare system in LMIC settings to address this issue [27]. Our study innovatively explores how chronic disease (namely, hypertension) is considered within the Peruvian primary healthcare system, highlighting how health systems could be strengthened to better address chronic disease.

Our results align with other research showing that global policies and funding are more often focused on communicable diseases and epidemic response, while hypertension is often deprioritized despite recognition of its increasing prevalence [28]. Another study in Peru also found that primary health care facilities were underfunded (especially for non-communicable diseases), and that surveillance systems were unable to accurately track the disease burden to justify higher budgets [29]. Our study demonstrates that challenges faced by the health system in dealing with chronic disease are similar to those faced for communicable diseases. For example, medication shortages and distribution challenges are common barriers to treatment adherence in LMICs [6]. Other studies have similarly found that a lack of culturally competent care and increased distance from healthcare facilities deter healthcare seeking behavior [30], and lack of coordination across different healthcare providers (i.e. due to inadequate clinical information systems or lack of integrated platforms) are common challenges to continuity of care [31]. Other work in Peru has also found that clinical guidelines were not adapted to the local context, healthcare workers lacked training on chronic diseases, and shortages of medication and appropriately calibrated devices were lacking across primary care facilities [29]. Previous research that has included patient perspectives on hypertension control have found factors such as affordability of medical care and medications, motivation and capacity to adopt healthy lifestyle changes, knowledge about the importance of controlling hypertension, and perceived acceptability of healthcare services are key barriers to hypertension management [7]. These factors did not arise in our study given our focus on healthcare providers and the health system.

Our findings underscore important healthcare disparities in Peru. First, rural populations of Peru experience greater limitations in access to healthcare compared to urban populations [14]. Our research, conducted in the rural area of Puno, highlighted such challenges with rural healthcare infrastructure, including challenges with the supply and distribution of medications, lack of equipment, and shortages of healthcare providers. Second, indigenous populations often face ethnic and cultural disparities in access to healthcare [32], as our study similarly found. Third, lower-income populations face socioeconomic barriers to accessing healthcare [33]. Our study underscores the further barriers that the low-income population of Puno face when attempting to access hypertension care. For example, lower-income populations are more affected by medication shortages given that they are unable to purchase medications from private pharmacies.

## Policy implications

The results of our rapid assessment highlight the need for culturally appropriate, feasible, and affordable interventions for hypertension prevention and management at the supra-individual level to be integrated into national policies and clinical guidelines. Our findings also suggest the need for greater investment both in raising awareness of hypertension and healthy lifestyles for prevention and early diagnosis (as lack of knowledge is a commonly reported barrier to hypertension detection and control [7]), as well as for improving management and treatment of the disease (as controlling hypertension has been shown to be highly cost-effective [34]). In addition to raising awareness, previous research in Peru has demonstrated the importance of removing barriers to healthy lifestyles (i.e. limited availability and high cost of healthy food, gendered barriers to engaging in sports, geographic distance to healthcare centers, and poor treatment by doctors) [35].

Also, the limited availability of health personnel and accompanying heavy work burden borne by the existing staff suggest that interventions that shift some responsibilities to health facility providers, such as CHWs, could be an effective way to improve healthcare in the area. Other research similarly finds that shifting tasks from physicians to non-physician healthcare workers can relieve workloads and improve patient outcomes [36]. However, to be effective, CHWs must be integrated into and recognized as a formal part of the health system, including adequate renumeration, training, supervision, and inclusion within national policies and guidelines such as the Practical Clinical Guide for the Management of Hypertension and the Technical Guide for the Diagnosis, Treatment, and Control of Hypertension [15, 16]. These guidelines must also be systematically tailored to the local context, accounting for cultural practices and resource availability.

The lack of hypertension training among health professionals, especially CHWs, also highlights the need to provide training on culturally sensitive hypertension care as part of regular capacity building and continuing education. Our results also highlighted the importance of incorporating health considerations into all social programs. Our finding that medication shortages are not necessarily due to a lack of supplies but often to distribution that does not account for facility-level needs suggests a need to redesign the drug distribution system to avoid problems of overstocking and understocking, which could be facilitated by more cohesive and integrated electronic health information systems. These recommendations could be integrated into Peru's efforts to scale up the WHO HEARTS initiative to improve hypertension care in primary healthcare settings [37].

The design of our formative research activities enabled local stakeholders–healthcare providers themselves–to actively participate in identifying challenges to hypertension care in the Puno healthcare system. In addition to enabling an understanding of key issues that would need to be addressed to improve care in the region, this engagement also motivated local healthcare professionals to work toward addressing the identified issues. As a result of our participatory formative research activities, participants noted that they better understood the gravity and importance of improving hypertension care and expressed willingness and enthusiasm to collaborate closely on design and execution of the ANDES trial to work towards that goal; several providers have already been integrated as advisors into the ANDES research team. Such community-driven, bottom-up processes of stakeholder engagement for collaboratively developed interventions have been previously demonstrated to result in more effective, equitable, legitimate, and sustainable interventions because of stakeholder ownership and buy-in [20, 38]. The collaboration could also serve to stimulate greater attention to and prioritization of hypertension as a key public health issue in the region. Knowledge and awareness gained by participants through their engagement in our research process could be

disseminated within local health networks, which could ultimately lead to increased political support, strengthening of health policies, and increased funding for non-communicable disease care.

## Strengths and limitations

The findings from our participatory workshop and in-depth interviews provide useful context and guidance for improving hypertension care in the region. Our use of both group discussions at the workshop as well as individual interviews enabled triangulation between different sources and methods. However, our study also has some limitations. Although we included a range of provider types from different types of facilities and areas of Puno, all participants were healthcare professionals. Perspectives of patients, community members, and community health workers must also be considered to obtain a more complete understanding of both supply- and demand-side facets of the healthcare system. Additionally, while the policy implications we developed are based in part on participant recommendations, our paper does not explicitly identify participant-developed solutions to the reported barriers. Furthermore, the workshop and interviews were conducted in late 2021 to early 2022 when healthcare systems were still focusing on COVID-19 and operating under pandemic policies. It is possible that operating procedures and priorities could change as the country progresses post-pandemic. Nonetheless, our findings highlight important challenges related to hypertension care that must be addressed to ensure availability and accessibility of quality healthcare to curb the increasing rates and negative consequences of hypertension in Peru.

## Conclusion

Healthcare providers in Puno, Peru reported systems-level barriers to hypertension care including limited funding, lack of cultural consideration in national guidelines, inadequate distribution and occasional shortages of medications and equipment, lack of personnel/high turnover, limited training opportunities, and incompatible health information systems. Insights gained from health providers on the healthcare system in Puno provide essential contextual information to inform development of organizational-level strategies necessary to improve provider and patient behaviors to achieve better hypertension care outcomes. For example, policy makers could revise national hypertension guidelines to consider local cultural beliefs and resource availability and integrate health considerations into all social programs. Medication distribution systems could be re-designed to prevent overstocking and understocking issues, which could be aided by more cohesive and integrated electronic health information systems. Our research also demonstrated that policy makers require evidence from cost-benefit analyses to inform and enable them to obtain and direct more financial resources toward hypertension prevention and care efforts. The findings highlight the importance of understanding and addressing organizational-level barriers to ensure success of provider- and patient-level hypertension interventions.

## Supporting information

**S1 Fig. Example theory of change for a hypertension control intervention used to guide workshop discussion.**
(TIFF)

**S2 Fig. Example theory of change with group comments (English translation of Fig 1).**
(TIFF)

**S1 Table. Characteristics of workshop participants.**
(PDF)

**S2 Table. Characteristics of in-depth interview participants.**
(PDF)

**S1 File. Semi-structured interview guide used to structure in-depth interviews with health professionals.**
(PDF)

## Acknowledgments

The authors wish to acknowledge workshop and interview participants, including healthcare system stakeholders (MINSA, DIRESA, and EsSalud) and teams at the participating public health centers, as well as the A.B. PRISMA field staff.

**Disclaimer:** The content is solely the responsibility of the authors and does not represent the policy of the National Heart, Lung, and Blood Institute, National Institutes of Health, United States (U.S.) Department of Health and Human Services, or the U.S. Government.

**Trial registration:** ClinicalTrials.gov, ID: NCT05524987, Addressing Hypertension and Diabetes through Community-Engaged Systems in Puno, Peru (ANDES study), prospectively registered on September 1, 2021.

## Author Contributions

**Conceptualization:** Kendra N. Williams, Lindsay J. Underhill, Armando Valdés-Velásquez, Laura K. Beres, Lisa de las Fuentes, Elvin H. Geng, William Checkley, Joel Gittelsohn, Victor G. Davila-Roman, Stella M. Hartinger-Peña.

**Formal analysis:** Kendra N. Williams, Janeth Tenorio-Mucha, Karina Campos-Blanco, Antonia Fuentes Herbozo.

**Funding acquisition:** Elvin H. Geng, William Checkley, Joel Gittelsohn, Victor G. Davila-Roman, Stella M. Hartinger-Peña.

**Investigation:** Kendra N. Williams, Janeth Tenorio-Mucha, Karina Campos-Blanco, Lindsay J. Underhill, Armando Valdés-Velásquez, Laura K. Beres, Lisa de las Fuentes, Lucy Cordova-Ascona, Zoila Vela-Clavo, Gonzalo Mariano Cuentas-Canal, Juan Carlos Mendoza-Velasquez, Sonia Mercedes Paredes-Barriga, Raquel Hurtado La Rosa, William Checkley, Victor G. Davila-Roman, Stella M. Hartinger-Peña.

**Methodology:** Kendra N. Williams, Lindsay J. Underhill, Armando Valdés-Velásquez, Laura K. Beres, Lisa de las Fuentes, Elvin H. Geng, William Checkley, Joel Gittelsohn, Victor G. Davila-Roman, Stella M. Hartinger-Peña.

**Project administration:** Janeth Tenorio-Mucha, Karina Campos-Blanco, Lucy Cordova-Ascona, Zoila Vela-Clavo.

**Resources:** Elvin H. Geng, William Checkley, Joel Gittelsohn, Victor G. Davila-Roman, Stella M. Hartinger-Peña.

**Supervision:** Makeda Williams, Elvin H. Geng, William Checkley, Joel Gittelsohn, Victor G. Davila-Roman, Stella M. Hartinger-Peña.

**Writing – original draft:** Kendra N. Williams.

**Writing – review & editing:** Janeth Tenorio-Mucha, Karina Campos-Blanco, Lindsay J. Underhill, Armando Valdés-Velásquez, Antonia Fuentes Herbozo, Laura K. Beres, Lisa de las Fuentes, Lucy Cordova-Ascona, Zoila Vela-Clavo, Gonzalo Mariano Cuentas-Canal, Juan Carlos Mendoza-Velasquez, Sonia Mercedes Paredes-Barriga, Raquel Hurtado La Rosa, Makeda Williams, Elvin H. Geng, William Checkley, Joel Gittelsohn, Victor G. Davila-Roman, Stella M. Hartinger-Peña.

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
