## [Decision Letter · Decision Letter 0]

7 Dec 2023

PGPH-D-23-01538

Health system barriers to hypertension care in Peru: Rapid assessment to inform organizational-level change

Dear Dr. Williams,

Thank you for submitting your manuscript to PLOS Global Public Health. After careful consideration, we feel that it has merit but does not fully meet PLOS Global Public Health’s publication criteria as it currently stands. Therefore, we invite you to submit a revised version of the manuscript that addresses the points raised during the review process.

The manuscript is generally well written, and presents several insights about health system barriers to hypertension care in Peru. However, I concur with the comments of both the referees, and believe that the manuscript will improve substantially if these comments can be addressed.

We look forward to receiving your revised manuscript.

Kind regards,

Sarthak Gaurav

Academic Editor

Journal Requirements:

Additional Editor Comments (if provided):

Reviewers' comments:

Reviewer's Responses to Questions

**Comments to the Author**

1. Does this manuscript meet PLOS Global Public Health’s publication criteria? Is the manuscript technically sound, and do the data support the conclusions? The manuscript must describe methodologically and ethically rigorous research with conclusions that are appropriately drawn based on the data presented.

Reviewer #1: Yes

Reviewer #2: Yes

2. Has the statistical analysis been performed appropriately and rigorously?

Reviewer #1: Yes

Reviewer #2: N/A

3. Have the authors made all data underlying the findings in their manuscript fully available (please refer to the Data Availability Statement at the start of the manuscript PDF file)?

Reviewer #1: Yes

Reviewer #2: Yes

4. Is the manuscript presented in an intelligible fashion and written in standard English?

Reviewer #1: Yes

Reviewer #2: Yes

5. Review Comments to the Author

Reviewer #1: Abstract and introduction

• Line 76-79: Reference

• Line 68- 85: It would be ideal if the flow of the content follows literature review globally, within LMIC, Latin America and then Peru

• Line 134-136: Could you please provide a rationale for provision of information for anaemia in children (<12 months old) for an article that focuses on hypertension

Methods

• Line- 166-167: Could you please provide a reference for the draft theories of change that the research team presented?

• Line 182-188: Could you please add a table for sampling strategy of all participants including no of each participant, type of interview, selection of participants etc.

• Line 201- 203: Was the inter-coder reliability checked in Atlas Ti.? Or were there any other methods used to check the reliability? How was the validity measured?

Results, discussion, conclusions

• Could you please give some policy-specific examples of how the study informs the current policies that addresses hypertension. (For. E.g., Peru’s National Health Strategy for Control and Prevention of Non-Communicable Diseases, 2004)

• Is there any role of private health care system? If yes, could you please explain

• Could you please provide limitations of the study?

Data and supporting information

• Line 157-159: Could you please add PAHO’s (WHO’s) Rapid health system review template to the annexures for reference? Were there any modifications to the template based on the Peru’s context?

• Line 189-190: Could you please add the semi-structured interview guide to annexures

• Could you please provide an English version of Figure 1. Example theory of change with group comments

Reviewer #2: I am grateful for the invitation to review the manuscript "Health system barriers to hypertension care in Peru: Rapid assessment to inform organizational-level change".

The study combines qualitative methodologies to offer a comprehensive and detailed look at the main barriers to hypertension care in a specific area of Peru at the organizational level. The study is clearly justified, uses an appropriate methodology to meet its objectives and is clearly written.

Therefore, most of my comments are suggestions to improve the text and its clarity.

METHODS

1. In the description of the "study setting" include information on poverty in Puno and how it ranks nationally.

2. In the description of the "Peruvian health care system" mention health insurance and its coverage in Peru. It would also be key to understand which services related to hypertension are covered by the insurance (i.e. care, medications, tests, etc.) and which are usually out-of-pocket expenses.

3. Any thoughts on the role of traditional medicine in the region?

4. This sentence from the description of the health system requires a reference: “After diagnosis, routine hypertension follow-up care can be provided at lower-level facilities, although some patients prefer to seek routine care from a cardiologist”.

5. If available, include the number of cardiologists in the Puno region, so that the reader can understand the situation of the health system and the scarce supply of specialists in Puno.

6. In the characterization of the participants in the workshop, add information about their roles (in addition to their profession). In the case of the interviews, you do detail if they have "leadership positions".

7. In "Data Analysis" describe the procedure for the generation of the (sub)codes within the 6 themes.

RESULTS

8. A general comment is that the results section, which is about 10 pages long, feels a bit long and could be shortened. It's not essential, so I leave that up to the editor.

9. Although the barriers identified are at the organizational level, I wondered if some were associated with gender (e.g., service hours may be a more problematic issue for men than for women?) and if there were any perceived differences between the two majority ethnic groups in Puno: the Aymara and the Quechua, regarding the barriers. If identified, it would be relevant to include them in the results.

10. In the table describing the barriers it does not make sense to include the following: "Integration of CHWs into the healthcare system could improve care". That seems to be more of an opportunity and I understand that it is more aligned -and perhaps also biased? - to questions the authors asked to design their intervention.

11. In the results section, when talking about social programs, it is not accurate to say that JUNTOS is not integrated with health goals. As far as I remember, it is a conditional transfer program, and one “condition” is the children's health. In any case, the sentence should be more precise.

DISCUSSION

12. Are there some barriers that do not appear in your study and do appear in other studies in similar / different settings? Are there barriers that were expected and did not appear in your study? Could the informants chosen have something to do with these absences (i.e. not having service users)?

13. In the introduction section, one of the objectives of the qualitative work is detailed as "to inform the design of an intervention to be tested in a randomized controlled trial". It would be interesting for one of the paragraphs of the discussion to reflect on the decisions that were derived from these results to design the intervention. This would allow us to reflect on the value of these formative studies.

14. The discussion states that "interventions that shift some responsibilities to health facility providers, such as CHWs, could be an effective way to improve healthcare in the area". But the results themselves indicate that their inclusion presents challenges. The discussion should reflect on mechanisms to overcome these challenges - not only their training but their inclusion within the system.

15. Two important absences -or that would have been an important contribution- are that the participants proposed measures to address the barriers, and having the viewpoint of decision-makers and users.

16. There are two ideas from the findings that I believe should be part of the discussion, in a subsection on policy implications: "For example, policy makers could revise national guidelines on hypertension to take into account local cultural beliefs and resource availability, and integrate health considerations into all social programs. Drug distribution systems could be redesigned to avoid problems of overstocking and understocking, which could be helped by more cohesive and integrated electronic health information systems."

6. PLOS authors have the option to publish the peer review history of their article (what does this mean?). If published, this will include your full peer review and any attached files.

**Do you want your identity to be public for this peer review?** For information about this choice, including consent withdrawal, please see our Privacy Policy.

Reviewer #1: No

Reviewer #2: **Yes: **Francisco Diez-Canseco

---

## [Decision Letter · Decision Letter 1]

8 Apr 2024

PGPH-D-23-01538R1

Health system barriers to hypertension care in Peru: Rapid assessment to inform organizational-level change

Dear Dr. Williams,

Thank you for submitting your manuscript to PLOS Global Public Health. After careful consideration, we feel that it has merit but does not fully meet PLOS Global Public Health’s publication criteria as it currently stands. Therefore, we invite you to submit a revised version of the manuscript that addresses the points raised during the review process.

Please note that we have only been able to secure a single reviewer to assess your manuscript. We are issuing a decision on your manuscript at this point to prevent further delays in the evaluation of your manuscript. Please be aware that the editor who handles your revised manuscript might find it necessary to invite additional reviewers to assess this work once the revised manuscript is submitted. However, we will aim to proceed on the basis of this single review if possible. 

The manuscript has been evaluated by one reviewer, and their comments are available below.

The reviewer has raised a number of minor concerns. They recommend improving the results section, specifically by discussing potential additional barriers, and discussing potential evidence for other unhealthy behaviours. They additionally recommend extending the discussion section, by discussing how the results fit in with the existing body of knowledge. Could you please carefully revise the manuscript to address all comments raised?

We look forward to receiving your revised manuscript.

Kind regards,

Johanna Pruller, Ph.D.

PLOS Staff Editor

Journal Requirements:

2. We have noticed that you have uploaded Supporting Information files, but you have not included a list of legends. Please add a full list of legends for your Supporting Information files after the references list.

Additional Editor Comments (if provided):

Reviewers' comments:

Reviewer's Responses to Questions

**Comments to the Author**

1. If the authors have adequately addressed your comments raised in a previous round of review and you feel that this manuscript is now acceptable for publication, you may indicate that here to bypass the “Comments to the Author” section, enter your conflict of interest statement in the “Confidential to Editor” section, and submit your "Accept" recommendation.

Reviewer #1: (No Response)

2. Does this manuscript meet PLOS Global Public Health’s publication criteria? Is the manuscript technically sound, and do the data support the conclusions? The manuscript must describe methodologically and ethically rigorous research with conclusions that are appropriately drawn based on the data presented.

Reviewer #1: (No Response)

3. Has the statistical analysis been performed appropriately and rigorously?

Reviewer #1: (No Response)

4. Have the authors made all data underlying the findings in their manuscript fully available (please refer to the Data Availability Statement at the start of the manuscript PDF file)?

Reviewer #1: (No Response)

5. Is the manuscript presented in an intelligible fashion and written in standard English?

Reviewer #1: (No Response)

6. Review Comments to the Author

Reviewer #1: Results:

• Not all key themes identified in Table 1 are addressed in the results section, potentially causing difficulty in following the results section. It may be helpful to specifically number and elaborate on each theme.

• The evidence presented under "National guidelines are incompatible with the local context" suggests only financial constraints, rather than cultural or language barriers, are the primary issue.

• While the study focuses on promoting a healthy diet for managing hypertension, it's unclear if there is evidence addressing other unhealthy behaviors such as smoking, alcohol consumption, physical inactivity.

• Regarding "Limited funding for hypertension care," does the study provide evidence of funding allocation specifically for hypertension awareness, considering participants' suggestions for improved database and tracking? Clarifying the significance of raising awareness and its potential impact could enhance the study's depth.

• What are the three different levels of the health system, and is the data used for decision-making at each level?

Discussion:

• The discussion section, being only one page, may be considered relatively short. The decision is left to the editor's discretion.

• How do the findings of this study contribute innovatively to the existing body of knowledge on health system assessments?

• Are the identified barriers to hypertension care in Puno inconsistent with the existing literature on healthcare systems in similar settings?

• How do the identified barriers align with the broader context of healthcare disparities and challenges in Peru?

7. PLOS authors have the option to publish the peer review history of their article (what does this mean?). If published, this will include your full peer review and any attached files.

**Do you want your identity to be public for this peer review?** For information about this choice, including consent withdrawal, please see our Privacy Policy.

Reviewer #1: No

---

## [Decision Letter · Decision Letter 2]

5 Jun 2024

PGPH-D-23-01538R2

Health system barriers to hypertension care in Peru: Rapid assessment to inform organizational-level change

Dear Dr. Williams,

Thank you for submitting your manuscript to PLOS Global Public Health. After careful consideration, we feel that it has merit but does not fully meet PLOS Global Public Health’s publication criteria as it currently stands. I was newly assigned this manuscript and recognize that the reviews and revisions to date have substantially improved it and have addressed the bulk of reviewers' concerns. I provide below remaining issues to address and anticipate a rapid conclusion to the review. Therefore, we invite you to submit a revised version of the manuscript that addresses the points raised during the review process.

We look forward to receiving your revised manuscript.

Kind regards,

Hannah Hogan Leslie, PhD

Academic Editor

Journal Requirements:

Additional Editor Comments (if provided):

- Line 73: clarify these %s are out of those living with hypertension, not all adults

- Line 114: add date of the data cited (and are these the most recent statistics prior to time of the study?). SIS eligibility has expanded (to the point that MINSA considers the full population to be covered, even while individuals may report lack of insurance), so it is important to contextualize this evidence.

- Line 129 - It may be more appropriate to say 'Blood pressure should be measured' in defining the guidelines, given gaps between guidelines and practice

- Lines 135 - 143: Reviewer 1 raised a concern around the inclusion of anemia here; I agree that the information on CHW roles (or lack of role) for hypertension is not a methodological component of the paper to the extent that it is presented here, and should be described more briefly, with elaboration in the Discussion section as desired.

- Line 222: "internal generalizability" - suggest "internal validity" as a more clear and familiar term

The revisions to the discussion section further strengthen this element of the manuscript; however, they rely primarily on review or overview articles across LMICs, with more limited engagement with what is already known and attempted for CVD control in Andean Peru. I would suggest further engagement with country or population-specific literature to enrich this section. Articles that touch on similar barriers to those identified in this manuscript include:

https://link.springer.com/article/10.1007/s11906-023-01286-w

https://www.tandfonline.com/doi/full/10.1080/16549716.2021.1975920

https://www.ssph-journal.org/journals/international-journal-of-public-health/articles/10.3389/ijph.2021.1604117/full

Reviewers' comments:

Reviewer's Responses to Questions

**Comments to the Author**

1. If the authors have adequately addressed your comments raised in a previous round of review and you feel that this manuscript is now acceptable for publication, you may indicate that here to bypass the “Comments to the Author” section, enter your conflict of interest statement in the “Confidential to Editor” section, and submit your "Accept" recommendation.

Reviewer #1: All comments have been addressed

2. Does this manuscript meet PLOS Global Public Health’s publication criteria? Is the manuscript technically sound, and do the data support the conclusions? The manuscript must describe methodologically and ethically rigorous research with conclusions that are appropriately drawn based on the data presented.

Reviewer #1: Yes

3. Has the statistical analysis been performed appropriately and rigorously?

Reviewer #1: Yes

4. Have the authors made all data underlying the findings in their manuscript fully available (please refer to the Data Availability Statement at the start of the manuscript PDF file)?

Reviewer #1: Yes

5. Is the manuscript presented in an intelligible fashion and written in standard English?

Reviewer #1: Yes

6. Review Comments to the Author

Reviewer #1: (No Response)

7. PLOS authors have the option to publish the peer review history of their article (what does this mean?). If published, this will include your full peer review and any attached files.

**Do you want your identity to be public for this peer review?** For information about this choice, including consent withdrawal, please see our Privacy Policy.

Reviewer #1: No

---

## [Editor Report · Decision Letter 3]

16 Jul 2024

Health system barriers to hypertension care in Peru: Rapid assessment to inform organizational-level change

PGPH-D-23-01538R3

Dear Dr. Williams,

We are pleased to inform you that your manuscript 'Health system barriers to hypertension care in Peru: Rapid assessment to inform organizational-level change' has been provisionally accepted for publication in PLOS Global Public Health.

Best regards,

Hannah Hogan Leslie, PhD

Academic Editor